

# High prevalence and risk factors of undernutrition in adult cancer patients at Hawassa University Hospital: a call for targeted interventions

Simret Girma Worku[1,*] and Zelalem Tafese Wondimagegne[2,*]

[1] School of Nutrition Food Science and Technology, Hawassa University, Hawassa, Sidama, Ethiopia
[2] School of Nutrition Food Science and Technology, Hawassa University, Hawassa, Sidama, Ethiopia
[*] These authors contributed equally to this work.

Corresponding author
Zelalem Tafese Wondimagegne,
wudasiez@gmail.com

## ABSTRACT

**Background.** Cancer, a condition marked by the uncontrolled growth of abnormal cells, remains a major global health concern and is the second most common cause of death worldwide. Both the disease itself and its treatments can negatively impact taste, smell, appetite, and nutrient absorption, increasing the risk of malnutrition. This study aimed to determine the prevalence and factors associated with undernutrition among adult cancer patients, with the goal of identifying key characteristics that could inform targeted interventions to address nutritional challenges in this population.

**Method.** A cross-sectional study involving 393 participants was conducted at a hospital using a convenient sampling method. Data collection took place over a two-month period from April to June 2023 through face-to-face interviews, utilizing a structured questionnaire. Quantitative data were gathered using both the questionnaire and the Patient-Generated Subjective Global Assessment short form (PG-SGA-SF). Data were analyzed with SPSS version 20, applying both bivariate and multivariate logistic regression analyses to identify associated factors. Variables with a $p$-value less than 0.05 were considered statistically significant predictors.

**Results.** Based on the PG-SGA-SF score, 58% of cancer patients were found to be malnourished. According to BMI measurements, 21% of participants were classified as underweight, while over 41% reported low dietary diversity, and nearly 78% were in advanced stages (III and IV) of cancer. Undernutrition showed significant associations ($p < 0.01$) with BMI below 18.5 kg/m² (AOR = 2.81, 95% CI [1.29–6.08]), poor dietary diversity (AOR = 4.54, 95% CI [2.41–8.53]), consumption of semisolid and liquid diets (AOR = 4.41, 95% CI [2.41–8.10]), presence of nausea (AOR = 10.71, 95% CI [5.48–20.94]), and constipation (AOR = 6.42, 95% CI [2.95–13.97]).

**Conclusion.** The findings of this study reveal a considerable burden of undernutrition among adult cancer patients, emphasizing the critical need for routine nutritional screening as part of cancer care. Early identification of malnutrition is essential to effectively manage associated symptoms, prevent nutrition-related complications, and improve treatment outcomes. Integrating comprehensive nutritional assessment and support into oncology services is recommended to enhance the quality of care and overall prognosis for cancer patients.

## INTRODUCTION

Cancer is a severe illness marked by the uncontrolled proliferation of abnormal cells within particular tissues or organs, posing a serious threat to normal bodily functions (*Saini et al., 2020*). These malignant cells can invade surrounding tissues and spread to other areas of the body, causing a range of symptoms and health complications depending on the location and stage of the disease (*American Cancer Society, 2019a*; *American Cancer Society, 2019b*). Globally, cancer holds a prominent place in public health concerns, currently ranking as the second leading cause of death after cardiovascular diseases. Studies from various regions, including Europe and North America, have indicated that approximately one in every two hundred people will develop cancer during their lifetime (*American Cancer Society, 2019b*; *Public Health Canada, 2018*; *Cancer trends report, 2023*). It is estimated that cancer accounts for one in every six deaths worldwide (*Mathur, Nain & Sharma, 2015*). An individual's risk of developing cancer is influenced by several factors, such as age, dietary and lifestyle habits, genetic predisposition, and environmental exposures (*Blackadar, 2016*).

Malnutrition remains a widespread issue among cancer patients in Ethiopia, with reported prevalence rates ranging from 48.1% to 58.4%, influenced by factors such as disease stage, loss of appetite, and gastrointestinal complications like diarrhea (*Muhamed et al., 2022*; *Gebremedhin, Cherie & Tolera, 2021*). In sub-Saharan Africa, the cancer burden is also considerable, with approximately 800,000 new cancer cases and 500,000 cancer-related deaths recorded in 2020. Breast cancer (129,400 cases among women) and cervical cancer (110,300 cases) together accounted for nearly 30% of all cancer diagnoses in both sexes (*Bray & Parkin, 2022*).

Projections by the World Health Organization (WHO) estimate that by 2035, there will be 24 million new cancer cases and 14.5 million related deaths globally each year (*WHO, 2020*). In sub-Saharan Africa specifically, cancer incidence is expected to rise by 85%, reaching about 1.28 million new cases and 970,000 deaths annually by 2030 (*Hamdi et al., 2021*).

Within Ethiopia, cancer accounts for roughly 5.8% of all national mortality according to the National Cancer Registry (*FMoH, 2015*). Data from Addis Ababa suggest approximately 64,000 new cancer cases are diagnosed annually, a number expected to increase as behaviors associated with economic transition—including smoking, substance use, poor dietary habits, and physical inactivity—become more widespread (*Misganaw et al., 2015*). A review of national cancer trends between 2010 and 2019 estimated 53,560 new cancer cases and 39,480 cancer-related deaths (*Awedew, Asefa & Belay, 2022*). Among these, breast cancer leads as the most common cause of cancer mortality, responsible for 22.6% of deaths, followed by cervical cancer (9.3%), leukemia (6.1%), non-Hodgkin's lymphoma (4.8%), and colon cancer (4.5%) (*Afework et al., 2022*). Collectively, these figures reflect cancer's growing impact as a serious and escalating public health concern in Ethiopia.

The relationship between nutrition and cancer has been acknowledged since the early 20th century. Both the disease itself and its treatments can negatively affect taste, smell, appetite, and the body's ability to consume or absorb necessary nutrients, often leading to malnutrition—a condition caused by insufficient intake of essential nutrients. Malnutrition in cancer patients contributes to fatigue, weakness, impaired immunity, and reduced capacity to tolerate treatment (*Muthike, Imungi & Muchemi, 2017*).

Various studies has highlighted the adverse consequences of malnutrition in individuals with cancer, including worsened health outcomes, reduced survival rates, and increased healthcare costs (*Arends et al., 2017a*; *Arends et al., 2017b*; *Van Cutsem & Arends, 2015*; *Van Tap et al., 2023*). It has been linked to impaired immune function, greater susceptibility to infections, increased psychological stress, and significant loss of body weight and muscle mass (*Martin, Senesse & Gioulbasanis, 2015*; *Nishikawa et al., 2021*).

For example, a study from Japan involving gastric cancer patients reported a markedly higher incidence of surgical site infections among malnourished patients compared to those with adequate nutritional status (36% *vs.* 14%, $P < 0.0001$) (*Arends et al., 2017a*; *Arends et al., 2017b*). Weight loss remains a primary sign of cancer cachexia—a complex, inflammation-driven wasting syndrome common in advanced cancer stages. Even modest weight loss (over 2.4%) at the time of diagnosis is associated with increased risks of complications and mortality (*Madeddu et al., 2018*). Severe pre-treatment weight loss in cancer patients can range from 7% to 57%, further contributing to poor clinical outcomes and higher complication rates (*Zhu, Wang & Gao, 2018*).

Malnutrition is the second most frequently diagnosed condition among cancer patients. This nutritional risk results from a complex and multifaceted process that goes beyond simple starvation, involving factors such as psychological distress, poor dietary intake, gastrointestinal disturbances, elevated energy demands, decreased physical activity, and metabolic alterations affecting different organs and tissues (*Martin, Senesse & Gioulbasanis, 2015*; *Muscaritoli, Lucia & Farcomeni, 2017*). These challenges are largely driven by disease-related metabolic changes, severe side effects from cancer treatments, and socioeconomic barriers (*Ferigollo & Bazzan, 2018*). Treatment-associated complications—including fatigue, dry mouth, nausea, constipation, and diarrhea—further interfere with a patient's ability to consume adequate food and fluids (*Milliron, Packel & Dychtwald, 2022*).

While certain groups of cancer patients are at higher risk for malnutrition, evidence suggests that many of these patients remain untreated for this condition (*Trujillo, Claghorn & Dixon, 2019*). A prospective observational study across 22 oncology centers in Italy reported that 15–20% of cancer patients were malnourished at the time of diagnosis, with rates climbing to 80–90% among those with advanced-stage disease (*Muscaritoli, Lucia & Farcomeni, 2017*). Moreover, it is estimated that 10–20% of cancer-related deaths are due to the effects of malnutrition rather than the cancer itself (*Arends et al., 2017a*; *Arends et al., 2017b*). These findings confirm that malnutrition is not only common but also a significant contributor to increased morbidity and mortality in this population (*Gu et al., 2019*).

Research indicates that malnourished cancer patients often endure longer hospital stays (*Li et al., 2018*; *Abrha et al., 2019*). In contrast, those who maintain a varied and balanced diet are better able to meet their nutritional needs (*Arimond et al., 2011*; *Labadarios, Steyn*

& Nel, 2011). A diverse diet has been associated with lower risks of disease recurrence and functional decline in cancer patients. Nutritional guidelines emphasize the importance of dietary variety to ensure adequate intake of micronutrients and energy, particularly for individuals at risk of deficiencies. Maintaining proper nutritional status can improve treatment outcomes, enhance immune function, and support recovery throughout cancer care (*Muthike, Imungi & Muchemi, 2017*; *Schwedhelm et al., 2016*; *Ravasco, 2019*; *Arends et al., 2017a*; *Arends et al., 2017b*; *Doyle et al., 2006*).

Given this evidence, early nutritional assessment is crucial for implementing timely and effective nutritional interventions, reducing complications, and improving patient survival rates (*Parsons et al., 2023*). Recognizing the central role of nutrition in cancer management, this study aims to provide updated and comprehensive insights into the nutritional challenges faced by cancer patients. Using structured questionnaires and the Patient-Generated Subjective Global Assessment short form (PG-SGA-SF) (*Naphat, Nuttapong & Somthawin, 2021*), this study also explores the key factors influencing the nutritional status of patients receiving care at the Hawassa University Comprehensive Specialized Hospital Cancer Treatment Center.

## MATERIAL AND METHODS

### Study setting, design and participants

This study was carried out at the Hawassa University Comprehensive Specialized Hospital Cancer Treatment Center in Hawassa, Ethiopia. It is the sole comprehensive specialized hospital in the region, offering cancer treatment and management services to over 18 million people across Sidama, Southern, and Central Ethiopia, as well as parts of the Oromia Region. Each year, more than 800 new cancer patients receive chemotherapy and outpatient care at this center. The study was an institutional-based cross-sectional study conducted from April to June 2023.

### Sample size determination

The required sample size for the study was calculated using a single population proportion formula, with a malnutrition prevalence of 58.4% from a previous study at Tikur Anbessa Specialized Hospital's oncology unit in Addis Ababa, Ethiopia (*Gebremedhin, Cherie & Tolera, 2021*). A 5% margin of error with a 95% confidence level was applied. After adjusting for a 5% nonresponse rate, the final sample size was set at 393, ensuring it was practical and feasible given the study's time constraints and participant availability. Due to the limited number of eligible cancer patients during the two-month study period, a convenience sampling method was used, selecting participants based on accessibility. All eligible patients at the outpatient department were consecutively included. Research assistants were trained to identify eligible participants in the outpatient care unit and administer the survey questionnaires. Patients who agreed to participate provided oral informed consent. To be eligible for the study, patients had to be diagnosed with cancer of any stage, be 18 years or older, and be receiving cancer chemotherapy at the outpatient unit of the Hawassa University Comprehensive Specialized Hospital Cancer Treatment Center.

Critically ill patients and those diagnosed within the last three months were excluded, as they were unlikely to have experienced significant nutritional changes.

## DATA COLLECTION METHODS AND VARIABLES

### Outcome variable

*Nutritional status of patients with cancer*

In this study, we employed the PG-SGA-SF because it provides a streamlined and effective method for screening malnutrition risk while maintaining assessment accuracy. This tool is composed of four key sections: (1) Body weight, (2) Food intake, (3) Symptoms impacting oral intake, and (4) Physical activity and functional status. Scores range from 0, indicating no issues, to 36, reflecting severe nutritional problems. The overall PG-SGA score is calculated by summing the scores of each component (*Jager-Wittenaar & Ottery, 2017*). Using maximally selected rank statistics, a cut-off score of five was established as the optimal threshold for identifying malnutrition. Patients scoring between 0 and 5 are categorized as well-nourished, while those with a score of five or higher are considered malnourished (*Naphat, Nuttapong & Somthawin, 2021*; *Zhang et al., 2021*).

*Predictor variables*

The predictor variables in this study encompassed sociodemographic factors such as age, gender, education level, monthly household income, BMI, and hemoglobin (Hgb) levels, along with lifestyle factors including alcohol use and smoking. Cancer-specific variables like cancer type, stage, and treatment modality, as well as nutrition-related factors such as dietary diversity, were also taken into account. Hemoglobin levels and cancer-related data were obtained from patient medical records, while sociodemographic information, alcohol consumption, smoking habits, and dietary diversity were collected through interviews. Initially, bivariate logistic regression was conducted to screen variables, with those showing a $p$-value $\leq 0.25$ included in a multivariable logistic regression analysis (*Hosmer, 2018*). A backward stepwise selection procedure was applied, starting with all eligible variables and systematically removing the least significant ones until only statistically significant predictors remained.

*Anthropometric measures*

Anthropometric measurements were utilized to calculate body mass index (BMI) by dividing weight in kilograms by the square of height in meters. Following WHO guidelines, adults are classified as underweight (BMI < 18.5), normal weight (18.5–24.9), overweight (25–29.9), or obese ($\geq$30). Body weight was measured using a Seca electronic scale equipped with a stadiometer for height assessment, with values recorded to the nearest 0.1 kg. Participants were asked to remove their shoes and any heavy clothing prior to weighing, and a single measurement was taken (*Lahner, 2018*). Height was measured using a stadiometer featuring a sliding headpiece and graduated scale. Participants stood barefoot, upright against the measuring rod, and height was recorded by licensed nurses to the nearest 0.5 cm.

### Dietary diversity

Dietary diversity scores were calculated by totaling the number of different food groups, out of 16, consumed by participants as reported in a 24-hour dietary recall. This recall captured all foods eaten throughout the day—during breakfast, lunch, and dinner—whether at home or elsewhere, while snacks were recorded as consumed before or after main meals. Participants received one point for each food group they consumed at least once during the 24-hour period, and zero if they did not consume any items from that group. The average individual dietary diversity score (IDDS) served as the cutoff point (*Kennedy, Ballard & Marrie Claude, 2013*; *FAO & FHI, 2016*). To minimize bias related to specific days, data collection accounted for weekends and market days. Foods were categorized into groups without applying any minimum weight thresholds.

### Ethics

Ethical approval and clearance were granted by the Hawassa University College of Medicine and Health Sciences Ethical Review Board (IRB) (Ref No/P/G/C/2001/15). Following ethical clearance, a formal letter was sent from the Hawassa University School of Nutrition, Food Science, and Technology to the Hospital Cancer Treatment Center, and all necessary permissions were obtained. Informed consent was acquired prior to participant enrollment, with a thorough explanation of the study's objectives and procedures provided. Participants were asked to confirm their understanding of this information before giving consent. The confidentiality of all information collected from participants was strictly maintained.

### Quality control

A two-day training session was conducted for all data collectors (professional healthcare providers), and the questionnaire was pre-tested at a different hospital in Hawassa city. The collected data were reviewed to ensure content validity and completeness.

### Data analysis

Data were entered into EPI-INFO version 3.5.1 statistical software and then exported to SPSS for Windows version 25 for further analysis (IBM Corp., Armonk, NY, USA). Categorical variables were summarized using frequencies and proportions, while measures of central tendency and dispersion were applied to continuous variables. Factors with a $p$-value $\leq 0.25$ in bivariate analysis were included in the logistic regression model (*Hosmer, 2018*). A significance level of 0.05 was set for the final model. Multicollinearity was assessed using the variance inflation factor (VIF).

## RESULTS

### Socio-demographic characteristics of cancer patients

A total of 393 participants completed the questionnaire, resulting in a 100% response rate. Among the respondents, 218 (55.5%) were female and 175 (44.5%) were male. The average age of the participants was 43 years, with a median age of 42, ranging from 18 to 80 years. Over two-thirds (77.2%) of the participants had at least a primary school education. Nearly 240 (60.8%) of the participants reported a monthly income of 1,500 ETB or higher (see Table 1).

**Table 1  Socio-demographic characteristics of adults with cancer at Hawassa University Comprehensive Specialized Cancer Treatment Center, 2023 ($n = 393$).**

| Variables | $n$ (%) |
|---|---|
| **Age** | |
| 18–24 | 18 (4.6) |
| 25–34 | 66 (16.8) |
| 35–44 | 140 (35.5) |
| 45–54 | 80 (20.3) |
| 55–64 | 57 (14.5) |
| >65 | 32 (8.1) |
| **Sex** | |
| Male | 175 (44.5) |
| Female | 218 (55.5) |
| **Region** | |
| Sidama | 157 (39.8) |
| Oromia | 153 (38.9) |
| SNNPR | 83 (21.1) |
| **Marital status** | |
| Single | 49 (12.5) |
| Married | 315 (80.2) |
| Divorced | 8 (2) |
| Widowed | 21 (5.3) |
| **Occupation** | |
| Farmer | 84 (21.4) |
| Merchant | 72 (18.3) |
| Government employee | 79 (20.1) |
| Self-employee | 25 (6.4) |
| Student | 24 (6.1) |
| Other | 109 (27.7) |
| **Average monthly income** | |
| <500 ETB | 96 (24.4) |
| 500–1,499 ETB | 58 (14.7) |
| 1,500–2,500 ETB | 136 (34.6) |
| ≥2,500 ETB | 103 (26.2) |
| **Educational status** | |
| No formal education | 90 (22.9) |
| Primary education | 102 (25.9) |
| Secondary education | 107 (27.2) |
| College and above 94 (23.9) | |

**Notes.**

1 USD = 59 ETB, SNNPR = Southern Nations, Nationalities and peoples Regional state.

## Lifestyle characteristics of patients with cancer

Our results showed varying levels of substance use among the study participants. Specifically, 27.2% reported consuming alcohol, while 6.4% and 14% indicated smoking and chewing khat habits, respectively (Fig. 1). Chewing chat and alcohol consumption can

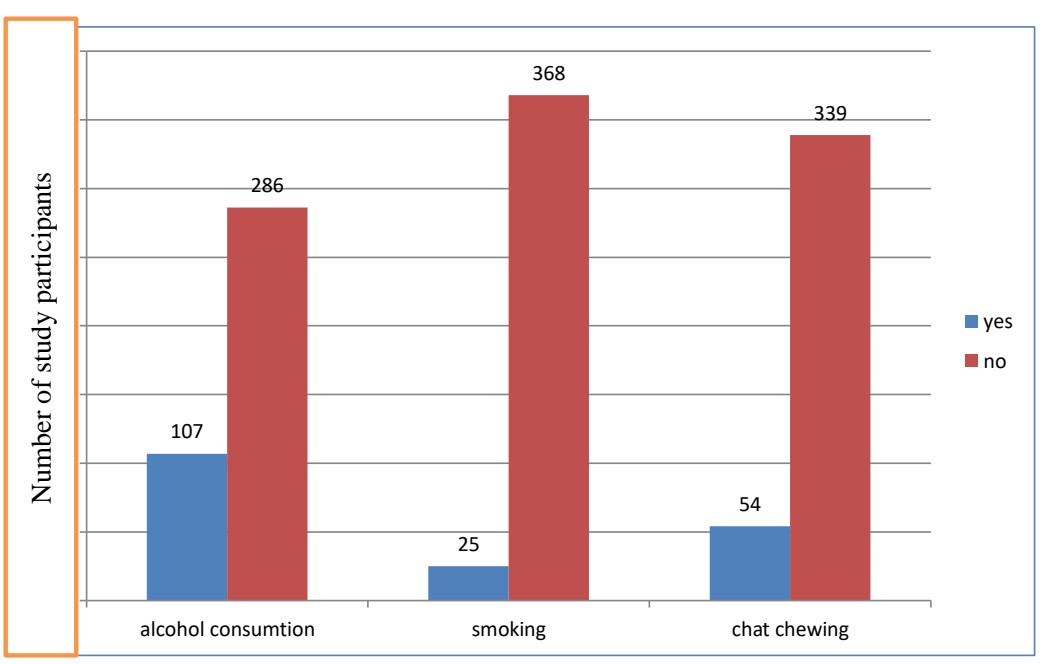

**Figure 1** Lifestyle characteristics of adults with cancer in Hawassa University Comprehensive Specialized Cancer Treatment Center, 2023 ($n = 393$).

significantly affect dietary intake by influencing appetite, nutrient absorption, and food choices. Chat chewing can suppress appetite, leading to reduced food intake, while alcohol consumption may result in poor dietary choices and nutrient loss from the body. Both behaviors can adversely affect overall nutrition.

### Disease and dietary characteristics of the respondents

The majority of respondents were in the advanced stages of their disease, with 35% in Stage III and 43% in Stage IV. Furthermore, approximately 44% of participants were only consuming semisolid and liquid food, 77.9% were in the advanced stages of cancer, and 33.1% had a hemoglobin level of ≤11 g/dl (Table 2).

### Characteristics of the respondents related to symptoms

Of the total participants, approximately 53% reported a loss of appetite, 7.6% experienced difficulty chewing or eating, 41.5% had suffered from nausea, 21% had experienced vomiting, 11% had dysphagia, and 20% had previously dealt with constipation (Table 3).

### Nutritional status of the respondents

The PG-SGA-SF results indicated that undernutrition is prevalent in approximately 58% of cases, with a higher occurrence in males than females. Regarding the distribution of BMI among participants, 21.4% were classified as underweight (BMI < 18.5 kg/m$^2$). 70.2% of participants fell within the normal weight range (18.5–24.9 kg/m$^2$), while 5.1% were classified as overweight (BMI 25–29.9 kg/m$^2$), and a relatively lower percentage 3.3% were obese (>30 kg/m$^2$) (Table 2).

**Table 2  Disease and anthropometric characteristics of adults with cancer in Hawassa University Comprehensive Specialized Cancer Treatment Center, 2023 ($N = 393$).**

| Variables | $n$ (%) |
|---|---|
| **Family history of cancer** | |
| Yes | 13 (3.3) |
| No | 380 (96.7) |
| **Comorbidity** | |
| Yes | 31 (7.9) |
| No | 362 (92.1) |
| **Stages of cancer** | |
| Stage I | 7 (1.8) |
| Stage II | 80 (20.4) |
| Stage III | 137 (34.9) |
| Stage IV | 169 (43.0) |
| **Hemoglobin level** | |
| HGB $\leq$ 11 g/dl | 130 (33.1%) |
| HGB $\geq$ 11 g/dl | 263 (66.9%) |
| **Types of cancer** | |
| Breast cancer | 116 (29.5) |
| Gastrointestinal cancer | 125 (31.8) |
| Sarcoma | 10 (2.5) |
| Lymphoma | 23 (5.9) |
| Gynecological cancer | 69 (17.5) |
| Lung cancer | 16 (4.07) |
| HENT cancer | 9 (2.3) |
| Hepatocellular cancer | 8 (2) |
| Others | 17 (4.3) |
| **Types of Treatment** | |
| Chemotherapy | 226 (57.5) |
| Chemo-surgery | 160 (40.7) |
| Chemo-radiation | 7 (1.8) |
| **Type of feeding** | |
| Solid | 219 (55.7) |
| Semi-solid | 151 (38.4) |
| Liquid | 23 (5.9) |
| **BMI** | |
| <18.5 kg/m$^2$ | 84 (21.4) |
| 18.5–24.9 kg/m$^2$ | 276 (70.2) |
| 25–29.9 kg/m$^2$ | 20 (5.1) |
| >30 kg/m$^2$ | 13 (3.3) |
| **PG-SGA –SF** | |
| Undernourished | 228 (58) |
| Well nourished | 165 (42) |

**Table 3 Symptoms related characteristics adults with cancer in Hawassa University Comprehensive Specialized Cancer Treatment Center, 2023 ($n = 393$).**

| Variables | $n$ (%) |
|---|---|
| **Chewing/eating problem** | |
| No | 363 (92.4) |
| Yes | 30 (7.6) |
| **Dysphagia** | |
| No | 350 (89.1) |
| Yes | 43 (10.9) |
| **Loss of appetite** | |
| No | 185 (47.1) |
| Yes | 208 (52.9) |
| **Feel nausea** | |
| No | 230 (58.5) |
| Yes | 163 (41.5) |
| **Diarrhea** | |
| No | 372 (94.7) |
| Yes | 21 (5.3) |
| **Vomiting** | |
| No | 311 (79.1) |
| Yes | 82 (20.9) |
| **Constipation** | |
| No | 315 (80.2) |
| Yes | 78 (19.8) |

## Dietary diversity of the respondents

Among the 393 participants, a significant proportion, 63.4%, had low dietary diversity scores, while 31.3% achieved medium dietary diversity, and only 5.3% attained high dietary diversity. This finding highlights a considerable dietary inadequacy among cancer patients in the study setting, indicating that most patients are at risk of micronutrient deficiencies due to limited variety in their daily food intake (Fig. 2).

## Types of cancer of the respondents

The majority of cancers reported by participants in this study were related to the digestive tract, including cancers of the esophagus, stomach, colon, rectum, and recto-sigmoid regions. Additionally, significant cancer types in the study population included breast cancer and gynecologic cancers, such as ovarian, cervical, and vulvar cancer (Table 2).

## Factors associated to the respondents' nutritional status

Bivariate logistic regression analysis was conducted to examine the association between various factors and undernutrition among cancer patients. Out of the ten selected variables that showed an association with undernutrition in the bivariate model (with $P < 0.25$), only five variables—BMI, dietary diversity, common type of diet, nausea, and a history of constipation—were found to significantly impact the nutritional status of cancer patients when analyzed using multivariate logistic regression (Table 4).
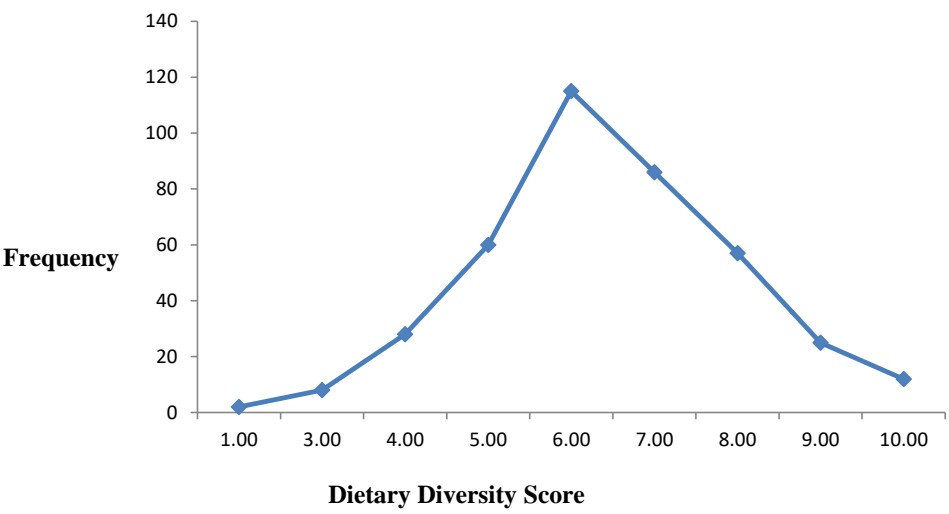

**Figure 2** Graph showing dietary diversity scores of adult cancer patients at Hawassa University Comprehensive Specialized Cancer Treatment Center, 2023 (*n* = 393).

Cancer patients with a BMI of less than 18.5 kg/m$^2$ had nearly three times the odds of undernutrition (adjusted odds ratio (AOR) = 2.81; 95% CI [1.29, 6.08]). Similarly, the odds of undernutrition were more than four times higher (AOR = 4.41; 95% CI [2.41, 8.10]) among those with low dietary diversity, and more than four times higher (AOR = 4.54; 95% CI [2.41, 8.53]) among those on semi-solid and liquid diets. Nausea was another significant factor, with undernutrition being more than 10 times higher (AOR = 10.71; 95% CI [5.48, 20.94]) in patients who reported nausea. Similarly, the odds of undernutrition were more than six times higher (AOR = 6.42; 95% CI [2.95, 13.97]) among cancer patients with a history of constipation compared to those without constipation.

## DISCUSSION

This study aimed to assess the nutritional status and related factors among adult cancer patients at the Hawassa University Comprehensive Specialized Hospital Cancer Treatment Center. The findings showed that more than half of the patients were undernourished, with prevalence rates similar to those reported in other Ethiopian studies and some international settings. For example, studies conducted in Addis Ababa hospitals found undernutrition rates of 58.4% and 57% (*Parsons et al., 2023*; *Adam et al., 2023*), and a Saudi Arabian study reported a prevalence of 51.1% (*Alsaleh et al., 2021*). However, the rate observed in this study was lower compared to reports from countries like South Korea (61%) (*Wie et al., 2010*), Iran (60.7%) (*Shahvazi, Onvani & Heydari, 2017*), China (77%) (*Ge, Lin & Yang, 2019*), Brazil (71.1%) (*Ferigollo et al., 2018*), and the Philippines (83%) (*Caballero, Lapitan & Buckley, 2013*). Conversely, our results were higher than those from West China Hospital (39.5%) (*Zhu, Wang & Gao, 2018*) and Egypt (33%) (*Abd Allah, Gad & Abdel-Aziz, 2020*). Such differences may arise from variations in population characteristics, sample sizes, sampling techniques, socioeconomic status, healthcare quality, or nutritional assessment tools.

**Table 4** Factors associated with nutritional status of adults (PG-SGA –SF) with cancer at Hawassa University Comprehensive Specialized Cancer Treatment Center, 2023 ($n = 393$).

| Variables | Undernutrition | | COR with 95% CI | AOR with 95% CI | P value |
|---|---|---|---|---|---|
| | Yes | No | | | |
| **Anemia** | | | | | |
| Yes | 36 | 11 | 1 | 1 | |
| No | 192 | 154 | 2.63 (1.29, 5.32)* | 1.65 (.58, 4.62) | 0.34 |
| **BMI** | | | | | |
| ≥18.5 kg/m² | 160 | 149 | 1 | 1 | |
| <18.5 kg/m² | 68 | 16 | 3.96 (2.20, 7.13)** | 2.81 (1.29, 6.08)** | 0.009 |
| **Dietary diversity** | | | | | |
| ≤5 | 136 | 25 | 1 | 1 | |
| >6 | 92 | 140 | 8.26 (5.02, 13.66)** | 4.54 (2.41, 8.53)** | 0.000 |
| **Common type of diet** | | | | 1 | |
| Solid | 82 | 137 | 1 | 1 | |
| Semi solid/liquid | 147 | 27 | 8.71 (5.34, 14.20)** | 4.41 (2.41, 8.10)** | 0.000 |
| **History of alcohol intake** | | | | | |
| Yes | 69 | 38 | 1 | 1 | |
| No | 159 | 127 | 1.45 (.91, 2.29) | 0.48 (.22, 1.06) | 0.07 |
| **Loss of appetite** | | | | | |
| Yes | 69 | 38 | 1 | 1 | |
| No | 37 | 109 | 5.34 (3.10, 9.22)** | 2.06 (.89, 4.78) | 0.090 |
| **Feel nausea** | | | | | |
| Yes | 147 | 16 | 1 | 1 | |
| No | 81 | 149 | 16.9 (9.44, 30.26)** | 10.71 (5.48, 20.94)** | 0.000 |
| **Educational status** | | | | | |
| College and above | 66 | 28 54 40 28 | 1 | 1 | 1 |
| Secondary school | 67 | 40 54 40 28 | .81 (.46, 1.43) | 0.67 (.28, 1.61) | 0.38 |
| Primary school | 48 | 54 | 1.53 (.86, 2.71) | 1.03 (.43, 2.45) | 0.93 |
| No formal education | 47 | 43 | 2.15 (1.18, 3.95)* | 1.38 (.56, 3.37) | 0.47 |
| **History of Chat chewing** | | | | | |
| Yes | 42 | 12 | 1 | 1 | 1 |
| No | 186 | 153 | 2.87 (1.46, 5.66)** | 2.15 (.88, 5.25) | 0.091 |
| **History of constipation** | | | | | |
| Yes | 65 | 13 | 1 | 1 | |
| No | 163 | 152 | 2.66 (2.47, 8.79)** | 6.42 (2.95, 13.97)** | 0.000 |

Notes.

*Statistically significant $P < .05$.

**Statistically significant $P < .001$.

[a] Reference categories.

AOR, adjusted odd ratio; accounts for potential confounding factors to estimate the independent effect of a predictor on the outcome; COR, crude odd ratio outcome between two groups without adjusting for other variables; BMI, body mass index.

In this study, low dietary diversity significantly influenced nutritional status. Cancer patients with lower dietary diversity scores were more prone to malnutrition, a pattern consistent with findings from Ethiopia and Iraq (*Fattahi et al., 2020*; *Girma & Nana, 2023*). Since dietary diversity reflects diet quality, patients consuming fewer food groups likely have inadequate nutrient intake, increasing their risk of undernutrition.

The study also identified semi-solid and liquid diets as major contributors to undernutrition, with 40.3% of patients unable to eat solid foods. Adequate nutrition is critical to managing treatment side effects and supporting immune function, yet many patients experience difficulties such as dysphagia and nausea. Similar studies report that problems with chewing and swallowing solid foods are common among cancer patients undergoing therapy (*Chen et al., 2018*). These issues can be aggravated by conditions like dry mouth, frequently observed in cancer patients (*Dysphagia Section Oral Care Study Group et al., 2012*), which reduces nutrient density intake and worsens malnutrition.

Additionally, 41.5% of patients reported nausea, which showed a strong association with undernutrition. Nausea and vomiting are well-known side effects of chemotherapy that significantly impair nutritional status (*Rao & Faso, 2012*; *Farrell et al., 2013*). This link aligns with other research showing that patients experiencing nausea and poor appetite are at elevated risk of malnutrition (*Conigliaro et al., 2020*; *Muhamed et al., 2022*), highlighting the importance of symptom management in nutritional care.

Constipation was also found to be a significant factor related to undernutrition in this patient population. Affecting nearly 60% of cancer patients (*Wickham, 2017*), constipation often results from cancer treatments, opioid use, surgeries, immobility, low-fiber diets, and dehydration. Consistent with prior studies (*FAO & FHI, 2016*; *Viana et al., 2020*), constipation was commonly reported and negatively impacted nutrition by reducing appetite due to discomfort and by impairing digestion and nutrient absorption. These results underline the necessity of effective constipation management to improve nutritional outcomes.

Overall, the findings emphasize the critical role of healthcare professionals in cancer treatment centers to identify and manage factors contributing to undernutrition. Personalized and comprehensive nutritional care is essential to optimize treatment success (*Gabor, Stein & Tommy, 2021*). Despite increasing recognition of the importance of nutrition in cancer care, research from Ethiopia and elsewhere shows that healthcare providers often lack adequate nutritional knowledge and practices (*Biruk et al., 2020*; *Alkhaldy, 2019*; *Munuo et al., 2016*; *Caldow, Palermo & Wilson, 2022*; *Tafese & Shele, 2015*). Therefore, the current quality of nutrition-related care in the study setting requires further evaluation through longitudinal and controlled studies.

The strength of this study lies in identifying malnutrition risk factors specific to cancer patients, highlighting the need for tailored regional and temporal interventions. However, its cross-sectional design limits the ability to establish cause-and-effect relationships.

## CONCLUSIONS

This study uncovers a notably high prevalence of undernutrition among adult cancer patients at the Hawassa University Comprehensive Specialized Cancer Treatment Center. Key contributors to undernutrition include nausea, reliance on semi-solid and liquid diets, limited dietary diversity, and constipation, reflecting the multifaceted nutritional challenges faced during cancer treatment. Compared to a previous study at the same center reporting a 48% malnutrition rate (*Muhamed et al., 2022*), our findings indicate an increasing trend. Additionally, the high prevalence of anemia underscores the necessity for proactive management of hematological complications alongside nutritional care.

The results emphasize the critical role of healthcare providers in addressing these nutritional issues through targeted interventions. Routine nutritional assessments should guide personalized care strategies, with nutrition integrated as a fundamental aspect of cancer treatment. Interventions should prioritize enhancing dietary diversity and nutrient intake while effectively managing symptoms like constipation and nausea through evidence-based methods. Furthermore, fostering collaboration among dietitians, nurses, and physicians is essential to deliver comprehensive nutritional support.

Given that healthcare professionals are at the forefront of patient care, strengthening their nutritional knowledge and skills within cancer care frameworks is crucial. Such capacity building can significantly improve the nutritional status, overall well-being, and treatment outcomes of cancer patients, ultimately contributing to more effective and holistic cancer management.

## ACKNOWLEDGEMENTS

We acknowledge the cancer patients who gave their time to participate in this study and the data collectors who tirelessly worked to complete this study. We are also grateful to the healthcare workers at the cancer treatment center of Hawassa Referral Hospital for their cooperation during data collection.

### Funding
The authors received no funding for this work.

### Competing Interests
The authors declare there are no competing interests.

### Author Contributions
- Simret Girma Worku conceived and designed the experiments, performed the experiments, analyzed the data, prepared figures and/or tables, authored or reviewed drafts of the article, and approved the final draft.
- Zelalem Tafese Wondimagegne conceived and designed the experiments, performed the experiments, analyzed the data, prepared figures and/or tables, authored or reviewed drafts of the article, and approved the final draft.

# PeerJ

## Human Ethics

The following information was supplied relating to ethical approvals (i.e., approving body and any reference numbers):

The Hawassa University, College of Medicine and Health Sciences, Ethical Review Board (IRB) approved the study (Ref No/P/G/C/2001/15).

## Data Availability

Raw data is available in the Supplemental Files.

## Supplemental Information

Supplemental information for this article can be found online at http://dx.doi.org/10.7717/peerj.19925#supplemental-information.

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
