# Peer review of "High prevalence and risk factors of undernutrition in adult cancer patients at Hawassa University Hospital: a call for targeted interventions"

_PeerJ, doi:10.7717/peerj.19925_

## Round 0.1 · original submission · Major Revisions

Dear Dr. Wondimagegne, Thank you for the opportunity to review this very interesting paper. Please note at this point that your manuscript needs major revisions, please be sure that your manuscript would benefit from extensive proofreading to improve clarity and ensure grammatical accuracy.
Also, please review and be more specific about the experimental design and pay attention to the reviewer's comments about the accuracy of your results.

**Language Note:** The Academic Editor has identified that the English language must be improved. PeerJ can provide language editing services - please contact us at [email protected] for pricing (be sure to provide your manuscript number and title). Alternatively, you should make your own arrangements to improve the language quality and provide details in your response letter. – PeerJ Staff

Reviewer 1 ·

Basic reporting

no comment

Experimental design

no comment

Validity of the findings

no comment

Additional comments

I commend the authors for their extensive data set. However, the manuscript lacks innovation, and the conclusions drawn from the research seem to have been known by everyone for a long time.

Reviewer 2 ·

Basic reporting

Thank you for the opportunity to review this manuscript. While the study addresses an important topic, several areas require attention before it can be considered for publication. My primary concerns are outlined below, with more detailed feedback available in the annotated manuscript:

• The manuscript would benefit from further proofreading to enhance clarity and ensure grammatical accuracy.

• Several paragraphs, particularly in the introduction and discussion, contain lengthy sentences with excessive information. Consider breaking these down into shorter, more focused sentences to improve readability.

Experimental design

• The cross-sectional design is appropriate for this study.

• There is inconsistency in the terminology used to reference the PG-SGA tool. Please clarify whether the full Patient-Generated Subjective Global Assessment (PG-SGA) or the Short Form (PG-SGA-SF) was utilized. If the Short Form was used, please provide a justification for this choice in the methods section.

• The inclusion and exclusion criteria require further clarification to ensure transparency and reproducibility.

Validity of the findings

The manuscript states that bivariate analyses were conducted prior to multivariate logistic regression. However, the presented results appear to be solely from the multivariate model. Please clarify whether bivariate analyses were performed and, if so, present those results separately before proceeding to the multivariate analysis. This would ensure the validity of the findings and improve the logical flow of the analysis.

The conclusion requires substantial revision. It should concisely summarize the key findings, their implications, and their contribution to the field. Avoid unnecessary repetition of information already presented in the discussion.

Additional comments

• The discussion provides valuable context through comparisons with other studies. However, it currently focuses excessively on specific prevalence rates from various countries. I recommend streamlining these comparisons to highlight key trends and regional variations, while emphasizing the core findings of the present study.

• The discussion on nausea, vomiting, and constipation is informative, but the level of detail on their physiological mechanisms would be too much.

Annotated reviews are not available for download in order to protect the identity of reviewers who chose to remain anonymous.

Reviewer 3 ·

Basic reporting

In general, the English language needs to be improved to ensure that readers can clearly understand your text. I would suggest polishing the language in this manuscript before it can be published.

• Line 27 Suggest updating "in April and June" to "from April to June" to avoid confusion and align with Line 142

• Line 36, need to explain the meaning of the acronym “AOR” as well as “COR” in the main text

• Line 47 The period at the end of the sentence is redundant.

• Line 54 Please double check the cancer incidence rate "one in two people"? From American Cancer Society, 2019b, for example, cancer incidence rate is one in three people in US population

• Line 82 What does the "(15-17)" refer to?

• Line 83 Missing commas in the first three symptoms

• Line 95 Please consider adding a comma between "simple starvation" and "including"

• Line 113-115: need a comma between “malnutrition” and “nutritional guidelines” in line 113. And “including cancer” should be rephrased to “including cancer patients”

• Line 109-110 and Line 117-119 please check the logic within these two sentences

• Line 128 "it is in debatable that" should be rephrased to "it is undebatable that".

• Line 129 the word "paramount" seems to be too strong

• Line 132 Please consider rephrasing the sentence to "factors associated with malnutrition in cancer patients"

• Line 141-144 The last sentence of this paragraph looks incomplete, you may want to add “from April to June” at the end of this sentence: “An institution-based cross-sectional study was conducted at Hawassa University Comprehensive Specialized Hospital Cancer Treatment Center.” And then remove the last sentence.

• Line 149 Suggest replacing "using 95% confidence level" with "with 95% confidence level".

• Line 173-175 Suggest updating as "participants’ demographic information (age, sex, education level, monthly household income, BMI, Hgb level, lifestyle, alcohol consumption, smoking), cancer-related information (type of cancer, stage of cancer, type of treatment), and nutrition related data (dietary diversity), instead of using "-".

• Line 220 Suggest changing the word "analyzed" to "summarized", as this is regarding summary tables/statistics of categorical and continuous variables, rather than statistical analysis with analytical models

• Line 253 please clarify why “however” was used for overweight group, do you expect a higher overweight percentage?

• Line 267-270 1) Please clarify whether the estimates in Table 4 are in log scale or not. 2) Please clarify in your logistic model, whether the outcome malnutrition is denoted as 1 or 0? 3) in Table 4, please update column names as "AOR with 95% CI", similar for COR

• Line 289 The percentage for Iran can be moved before the reference, to be consistent with the data for other countries.

• Line 290 Consider adding parentheses for " (77%) (Maria S, 2019), (71.1%) (De Melo Silva FR., etal 2015)", i.e. ((77%) (Maria S, 2019), (71.1%) (De Melo Silva FR., etal 2015))

• Line 348 "Notably, a study conducted two…" Is there a reference for this?

• Line 31, Line 35, Line 224 it looks that the significance levels for predictors are inconsistent.

Experimental design

Sampling technique:
• Line 150 Please clarify on how the “consecutive sampling technique” was done or add a reference. And in line 25, please clarify the difference between “convenient sampling technique” and “consecutive sampling technique”

Sample size calculation:
• Line 146 please clarify the single population proportion formula that is used in the calculation. In line 152, please clarify how the nonresponse rate was adjusted? Based on my calculation, the required sample size is (1.96^2*0.584*(1-0.584))/0.05^2=373, which is different from 393.

Statistical analysis:
• Line 176 Please clarify the bivariate analysis, is it the same as logistic regression?

• Line 176-177 please provide a reference to justify your variable selection method. The bivariate analysis used in this analysis, which keeps one factor in a model at one time and includes that factor in the final multivariable logistic model if the associated p value is less than or equal to 0.25, tends to ignore the potential correlation between predictors during the variable selection stage, e.g. education level and monthly household income may be correlated. Alternatively, you may consider some commonly used methods for selecting variables in logistic regression including forward selection, backward elimination, stepwise selection etc.

• Need to clarify in the main text how crude odds ratio and adjusted odds ratio are calculated, for the audience’s understanding

Validity of the findings

• Line 39-41 As your data were collected from a single center, how do your findings generalize to the entire or a larger population, such as patients in Ethiopia? You may want to rephrase your conclusion here or justify the generalizability.

• Line 250 Please justify the finding "more common in males than in females", e.g. by adding a sub-category of gender under PG-SGA –SF in Table 2

• Line 288 Missing the percentage for Korea

• Truncated words in Figure 3 x-axis

• Line 301 From Table 2, 55.7% (219) patients were able to consume solid foods. In table 4, however, the frequency of Semi solid/liquid is 137+82=219. Please double check the number to ensure consistency.

• Line 322 Constipation is in almost 60% cancer patients. Please clarify why constipation only accounts for 19.8% in your data (Table 3)?

• Line 350: In this sentence "elevated prevalence of anemia underscores…", 1) is this to compare anemia % in this study vs another study? If yes, please provide the data reference. 2) As anemia is not a significant factor based on your result, you may need to justify your statement on why it underscores the continuous overseeing requirement (e.g. a test comparing two proportions/prevalence)

---

## Round 0.2 · Minor Revisions

Dear Authors,

Thank you for revising your manuscript. The reviewers have made some minor suggestions to improve your work. Please review all the suggestions before approving your manuscript.

Reviewer 2 ·

Basic reporting

The introduction could be more concise. For example, one paragraph could be streamlined to focus directly on the implications of malnutrition among cancer patients, which would better align with the study’s objectives.

Experimental design

In the section discussing DDS and Figure 2, it would be helpful for the authors to clearly present the score categories (e.g., more than six food groups indicating high dietary diversity) to improve reader understanding.

Validity of the findings

Based on the data presented, the findings may only indicate a potential role in undernutrition. The interpretation should remain cautious to avoid overstating the results.

Additional comments

Thank you for the opportunity to review this manuscript. The study addresses a relevant topic; however, several areas require clarification, refinement, and alignment with the stated research objectives.

I have provided my detailed comments and suggestions in the attached file for the authors’ consideration. These remarks cover aspects related to the introduction, methodology, results interpretation, and overall coherence of the manuscript.

Annotated reviews are not available for download in order to protect the identity of reviewers who chose to remain anonymous.

Reviewer 3 ·

Basic reporting

No additional comments

Experimental design

No additional comments

Validity of the findings

No additional comments

---

## Round 0.3 · Minor Revisions

Thanks for the reviewed version of the manuscript.
However, there are some issues that should be addressed before acceptance.

Please review the following comments:

Title. Authors should consider eliminate “..in cancer patients” are repetitive.

Title. This is an institution-based study please consider adding “central Ethiopia”, or “Cancer treatment center” or Hawasa Universitary Hospital” to the title.

L35. Please consider eliminate “little” from “little over 41%..” if you have the exact data.

L41. Please consider “This report” instead “The report”.

L43 Eliminate PG-SGA-SFPG-SF

L43 Please change “Key Words” to “Keywords” and move it to next line.
The PG-SGA-SF are not described or referenced anywhere in the introduction section.

L163-170 There are five times “PG-SGA-SF” please, avoid repetitive sentences.

L186-195. Please re-write.

L195 Please move subtitle to next line

L243. BMI are reported twice in line 243 and 252. Please eliminate duplicate results

L251-255 The sum of the percentages of participants (BMI) do not compute 100%. (21.4%, 72%, 5.1%)

L261 Please move subtitle to next line.

L303 “Our findings identified semi-solid and liquid diets as a major contributor to undernutrition” please re-think this. ¿Does the hospital have nutritionist or some nutrition professional personnel?

Figure 1. Please add the scale of Y axis.

Figure 2. Add a better description of the graph.

Table 1. Please define SNNPR

Table 2. What does “Family history” means?

Table 2. Please make all the tables consistent.

Please, consider using a professional service to improve English.

**Language Note:** The Academic Editor has identified that the English language must be improved. PeerJ can provide language editing services - please contact us at [email protected] for pricing (be sure to provide your manuscript number and title). Alternatively, you should make your own arrangements to improve the language quality and provide details in your response letter. – PeerJ Staff

---

## Round 0.4 · accepted · Accept

We would like to express our gratitude to the authors for their efforts. The reviewer has thoroughly reviewed the revisions and is satisfied with the changes made. After careful review, I am pleased to inform you that the manuscript now meets the required standards. Therefore, I approve it for publication in its current form.

Reviewer 2 ·

Basic reporting

Thank you for the opportunity to review the amended manuscript. I have carefully examined the revisions and am satisfied with the changes made. The manuscript now meets the required standards, and I approve it for publication in its current form.

Experimental design

No further comments.

Validity of the findings

No further comments.